# Kuura—An automated workflow for analyzing WES and WGS data

**Dhanaprakash Jambulingam**[1], **Venkat Subramaniam Rathinakannan**[1], **Samuel Heron**[1], **Johanna Schleutker**[1,2], **Vidal Fey**[1,3] *

1 Institute of Biomedicine, Cancer Research Unit and FICAN West Cancer Centre, University of Turku and Turku University Hospital, Turku, Finland, 2 Department of Genomics, Laboratory Division, Turku University Hospital, Turku, Finland, 3 Faculty of Medicine and Health Technology/BioMediTech, Tampere University, Tampere, Finland

* vidal.fey@utu.fi

**Data Availability Statement:** Code Availability and Accessibility: The Kuura pipeline is currently hosted at the github repository https://github.com/dhanaprakashj/kuura_pipeline. Detailed information on installation and usage of the pipeline is provided in the attached supplementary files. Data

## Abstract

The advent of high-throughput sequencing technologies has revolutionized the field of genomic sciences by cutting down the cost and time associated with standard sequencing methods. This advancement has not only provided the research community with an abundance of data but has also presented the challenge of analyzing it. The paramount challenge in analyzing the copious amount of data is in using the optimal resources in terms of available tools. To address this research gap, we propose "Kuura—An automated workflow for analyzing WES and WGS data", which is optimized for both whole exome and whole genome sequencing data. This workflow is based on the *nextflow* pipeline scripting language and uses *docker* to manage and deploy the workflow. The workflow consists of four analysis stages—quality control, mapping to reference genome & quality score recalibration, variant calling & variant recalibration and variant consensus & annotation. An important feature of the DNA-seq workflow is that it uses the combination of multiple variant callers (*GATK Haplotypecaller*, *DeepVariant*, *VarScan2*, *Freebayes* and *Strelka2*), generating a list of high-confidence variants in a consensus call file. The workflow is flexible as it integrates the fragmented tools and can be easily extended by adding or updating tools or amending the parameters list. The use of a single parameters file enhances reproducibility of the results. The ease of deployment and usage of the workflow further increases computational reproducibility providing researchers with a standardized tool for the variant calling step in different projects. The source code, instructions for installation and use of the tool are publicly available at our github repository https://github.com/dhanaprakashj/kuura_pipeline.

## Introduction

Next generation sequencing technologies are fast becoming an important avenue for new discoveries in human genome research. They enable, for example, rapid identification of novel/rare variants amongst various populations, which in turn can help understanding the genetics behind particular diseases. The process of identifying variants from DNA-seq data is a multi-

availability: The publicly available datasets used in the evaluation are listed in Table 1 and were downloaded from the EBI FTP server: HG001: Exome sequencing of Homo sapiens: HG001 with Illumina NovaSeq 6000 IDT capture - SRR14724473 ftp://ftp.sra.ebi.ac.uk/vol1/fastq/SRR147/073/SRR14724473/SRR14724473_1.fastq.gz ftp://ftp.sra.ebi.ac.uk/vol1/fastq/SRR147/073/SRR14724473/SRR14724473_2.fastq.gz HG002: Exome sequencing of Homo sapiens: HG002 with Illumina NovaSeq 6000 IDT capture - SRR14724472 ftp://ftp.sra.ebi.ac.uk/vol1/fastq/SRR147/072/SRR14724472/SRR14724472_1.fastq.gz ftp://ftp.sra.ebi.ac.uk/vol1/fastq/SRR147/072/SRR14724472/SRR14724472_2.fastq.gz HG005: Exome sequencing of Homo sapiens: HG005 with Illumina NovaSeq 6000 IDT capture - SRR14724469 ftp://ftp.sra.ebi.ac.uk/vol1/fastq/SRR147/069/SRR14724469/SRR14724469_1.fastq.gz ftp://ftp.sra.ebi.ac.uk/vol1/fastq/SRR147/069/SRR14724469/SRR14724469_2.fastq.gz The IDT capture bed file used for coverage analysis and hap.py validation was obtained from: https://storage.googleapis.com/deepvariant/exome-case-study-testdata/idt_capture_novogene.grch38.bed.

**Funding:** The study was funded by grants to JS from Cancer Foundation Finland sr (https://syopasaatio.fi/en), grant #180146, and Jane and Aatos Erkko Foundation (https://jaes.fi/en/). The funders had no role in study design, data collection and analysis, decision to publish, or preparation of the manuscript.

**Competing interests:** The authors have declared that no competing interests exist.

step process involving numerous bioinformatics tools. The simplest way to run the analysis would be to select the right tools for each stage of the analysis and run them in a sequential manner; however, the major drawback of this approach is that this is 1. not portable, 2. not scalable, 3. not standardized in terms of software versions and 4. there are no checkpoints, thereby eliminating the reproducibility aspect of the analysis.

Multiple pipelines such as SeqMule [1], DNAp [2], Sarek [3], SpeedSeq [4], GESALL [5] and many more have been developed over the years to address the above-mentioned computational bottlenecks. However, they are either too rigid or too complex in their approach, leaving the user with little to no room for customizing the pipeline to their needs. Multiple frameworks such as *nextflow* [6], snakemake [7], cwltools, Pegasus [8], galaxy [9], and chipster [10], to name only a few, have been developed to enable users to create their own pipelines consisting of multiple processes. One pipeline that has been developed using the *nextflow* scripting language is Sarek. While this pipeline is easier to install and deploy than other frameworks with comparable functionality its comprehensive feature set makes it too complex for most intended users, limiting their ability to understand and modify the workflow.

We have developed the Kuura pipeline using *nextflow* framework to provide an end-to-end analysis pipeline from fastq files to an annotated consensus file. The pipeline makes use of recent state-of-the-art tools to increase both confidence and reproducibility of the generated results. The pipeline uses a combination of multiple variant callers—*GATK Haplotypecaller* [11], *DeepVariant* [12], *VarScan*2 [13], *Freebayes* [14] and *Strelka*2 [15]—and presents users with a consensus output which is then annotated. The pipeline uses *docker* [16] to manage and deploy the run environment.

## Results

We developed an automated analysis pipeline for sequencing data based on the need to routinely analyse large WGS and WES data sets produced at our in-house laboratory. To arrive at a versatilely applicable, standardized workflow the pipeline has been extensively tested and validated with publicly available gold standard datasets from the Genome in a Bottle Consortium (GIAB) hosted by the National Institute of Standards and Technology.

### Validation

The pipeline was validated against the gold standard WES datasets obtained from GIAB (see section Data Availability), outlined in Table 1, using the Haplotype comparison tool *hap.py* [17]. The results of the analysis are provided in Table 2.

### Discussion

We have developed Kuura, a sequence analysis workflow that uses *nextflow* & *docker* for performing reproducible end-to-end variant calling with minimal setup. Since *nextflow* and *docker* support POSIX compatible systems, Kuura can be easily deployed on all major operating systems such as Linux, MacOS and Windows. It supports cluster computing by means of Nextflow's integration with workload managers such as slurm, OAR, Sun Grid Engine (SGE)

**Table 1. Information on the gold standard datasets used for validating the pipeline.**

| SAMPLE ID | NIST ID | Pedigree | Project |
|---|---|---|---|
| NA12878 | HG001 | CEPH/UTAH | HapMap |
| NA24385 | HG002 | Ashkenazi Jewish | Personal Genome Project |
| NA24631 | HG005 | Han Chinese ancestry | |

**Table 2. Validation results using each variant caller.** The table shows the number of variants identified by each variant caller, their precision and recall values. *The table contains only SNP information.

| Variant Caller | Sample | | | | | | | | |
|---|---|---|---|---|---|---|---|---|---|
| | NA12878 | | | NA24385 | | | NA24631 | | |
| | Count | Precision | Recall | Count | Precision | Recall | Count | Precision | Recall |
| *GATK Haplotypecaller* | 160713 | 0.996402 | 0.958233 | 155973 | 0.995281 | 0.954474 | 150884 | 0.995416 | 0.951765 |
| *DeepVariant* | 261796 | 0.998768 | 0.989619 | 252227 | 0.999005 | 0.98792 | 247151 | 0.999355 | 0.987613 |
| *Freebayes* | 353434 | 0.94479 | 0.988765 | 342915 | 0.937996 | 0.986976 | 339658 | 0.948099 | 0.987294 |
| *Strelka*2 | 333574 | 0.916016 | 0.210014 | 317965 | 0.907925 | 0.214567 | 307473 | 0.908015 | 0.229228 |
| *VarScan*2 | 95416 | 0.982484 | 0.972359 | 94501 | 0.977653 | 0.970882 | 92490 | 0.983325 | 0.972397 |

amongst others, help in parallelising the analysis steps, which enables the users to scale up their workflows with minimal configuration changes. Since the entire analysis environment is *dockerized* and available from dockerhub it is highly portable and, at the same time, ensures reproducible results. For experienced users, it is possible to upgrade any component or the entire workflow. We recommend re-validating the pipeline after every change to any tool or other source code component to ensure the consistency of the results. The steps for a minimal validation run are described in the documentation.

In its current state, the pipeline requires a bed file for calculating and visualizing WES coverage information that is usually included with the sequencing data. The *mosdepth* tool [18] used for calculating coverage performs also without a bed file and does accept region limits via other tool parameters but for simplicity reasons Kuura demands the bed file when specifying WES data as input. If no bed file is available to the user one solution is to set the input flag to WGS which will take more computing time but arrive at the same output. Alternatively, users can create a bed file based on the input data as described in the FAQ section of the github repository.

The validation of the pipeline has been done using GIAB gold standard datasets listed in Table 1 using the Haplotype Comparison Tools (hap.py) to calculate the recall and precision values as follows:

Recall = $TP/(TP+FN)$

Precision = $TP/(TP+FP)$

where true positives (TP), false negatives (FN) and false positives (FP) are determined based on the input truth set. The implementation of the validation follows the procedure published by the DeepVariant authors (https://github.com/google/deepvariant/blob/r1.5/docs/deepvariant-training-case-study.md).

The results of analysis of the three datasets are provided in Table 2. The most obvious difference between the tools is their varying number of called variants. This behaviour was expected as all tools come with different inherent variant calling algorithm, default settings and filters. For example, VarScan2 has identified the smallest number of variants a behaviour which has previously been reported in various studies comparing the performance of different variant callers (e.g., [19–21]). At the same time, VarScan2 shows high precision (the highest number of true positives among all positions identified as variants) and recall (the total correctly identified positive among all identifiable variants in the sample), also corresponding to the earlier findings. Strelka2, on the other hand, shows the lowest performance of all callers employed in Kuura, with moderate precision values and low recall values. This reduced performance is likely a result of the relatively low number of mutations and only moderately high sequencing depth (~50X on average) based on a study comparing variant caller performance while systematically varying sequencing depth and mutation frequency [22].

By generating a consensus call set, Kuura, by default, produces a high-confidence variant list giving users consistent results to be used, e.g., in a standard detection variant workflow. The low-confidence set is kept, i.e., the output of the individual callers is available if needed.

Compared to other DNA-seq pipelines, Kuura is not feature intensive and it is not meant to be. It was designed to be an end-to-end data analysis pipeline with a user-friendly implementation generating comprehensive yet reliable, ready-to-use output for downstream analysis.

## Methods

The Kuura pipeline was constructed exclusively for the purpose of analyzing whole genome sequencing data and whole exome sequencing data. The pipeline has been categorized into four stages based on their functionalities; an illustration showing the stages is provided in Fig 1 and a description of each stage is explained below.

**1. Stage 1**:

### Quality control

NGS data cannot be used directly as they might contain sequencing bias in individual base composition, low quality reads, contamination from adapter sequences and overrepresented sequences. The raw fastq files are preprocessed in order to have their quality assessed and to check/remove any of the abovementioned anomalies from the data. Quality control has been implemented in the pipeline using the tools–*FASTQC* [23], fastp [24] and cutadapt [25]. *FASTQC* tool checks the fastq files and provides a multitude of information about the sequences including, but not limited to, per sequence GC content, overrepresented sequences, or adapter content. After *FASTQC* analysis, the reads are trimmed using cutadapt if the adaptor sequences are defined in the configuration file, else using fastp. After quality trimming, *FASTQC* is executed on all pre-processed fastq files for validation purposes.

**2. Stage 2**:

### Genome alignment and quality score recalibration

Following quality control, the preprocessed reads are aligned to the reference genome which is GRCh38 in the current version of the pipeline. The reads are aligned using the BWA-MEM software and the unsorted SAM (Sequence Alignment Map) output files generated by BWA are coordinate-sorted using *SAMtools* and compressed to Binary Alignment Map (BAM) files. In the next step duplicate reads are tagged using *MarkDuplicates*. After duplicate removal, *SAMtools* is used again to sort the BAM files based on their coordinates and index them. In the final step of stage 2, the base quality scores of the sorted and indexed reads are adjusted by the *GATK* Base Quality Score Recalibration (BQSR) tool, a machine learning algorithm detecting systematic scoring errors introduced by the sequencer.

**3. Stage 3**:

### Variant calling & variant recalibration

After adjusting the base quality scores, the next step in the analysis process is identifying variants in the sample. The pipeline incorporates 5 different variant callers—*DeepVariant*, *GATK Haplotypecaller*, *Freebayes*, *Strelka*2 and *VarScan*2. After variant calling, scores of the variants identified by *GATK Haplotypecaller* are recalibrated for SNPs and INDELs to filter out remaining artifacts from the dataset.

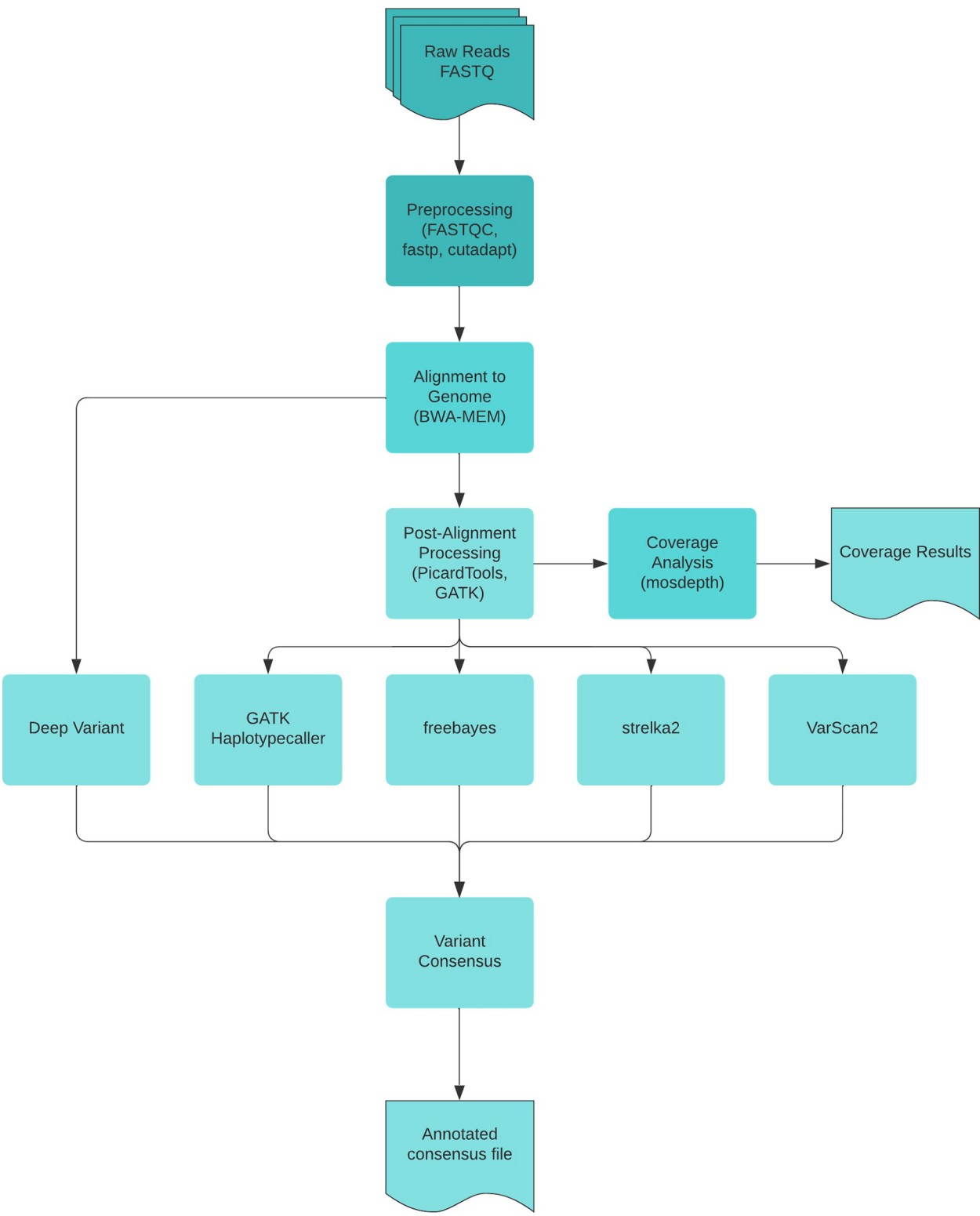

**Fig 1. Summary of the steps executed by the Kuura pipeline.**

**4. Stage 4**:

### Variant consensus & annotation

All five variant callers are applied to each sample and a 'high confidence' set is produced intersecting the results of all callers. Remaining variants, identified by only one method, are then treated as an additional 'low confidence' variant set. This approach prioritizes precision over recall in order to reduce the possibility of false positive variant calls. After identifying variants, it is important to functionally characterize and integrate predictive information about them. Functional annotation has been implemented in the pipeline using the Variant Effect Predictor (*VEP*) [26]. The annotation is applied only to the 'high confidence' consensus file.

### 5. Summary

All output, including QC, log files and compatible textual output of the individual processing tools, is summarized in a single report using *MultiQC* [27] and saved to a user-defined directory.

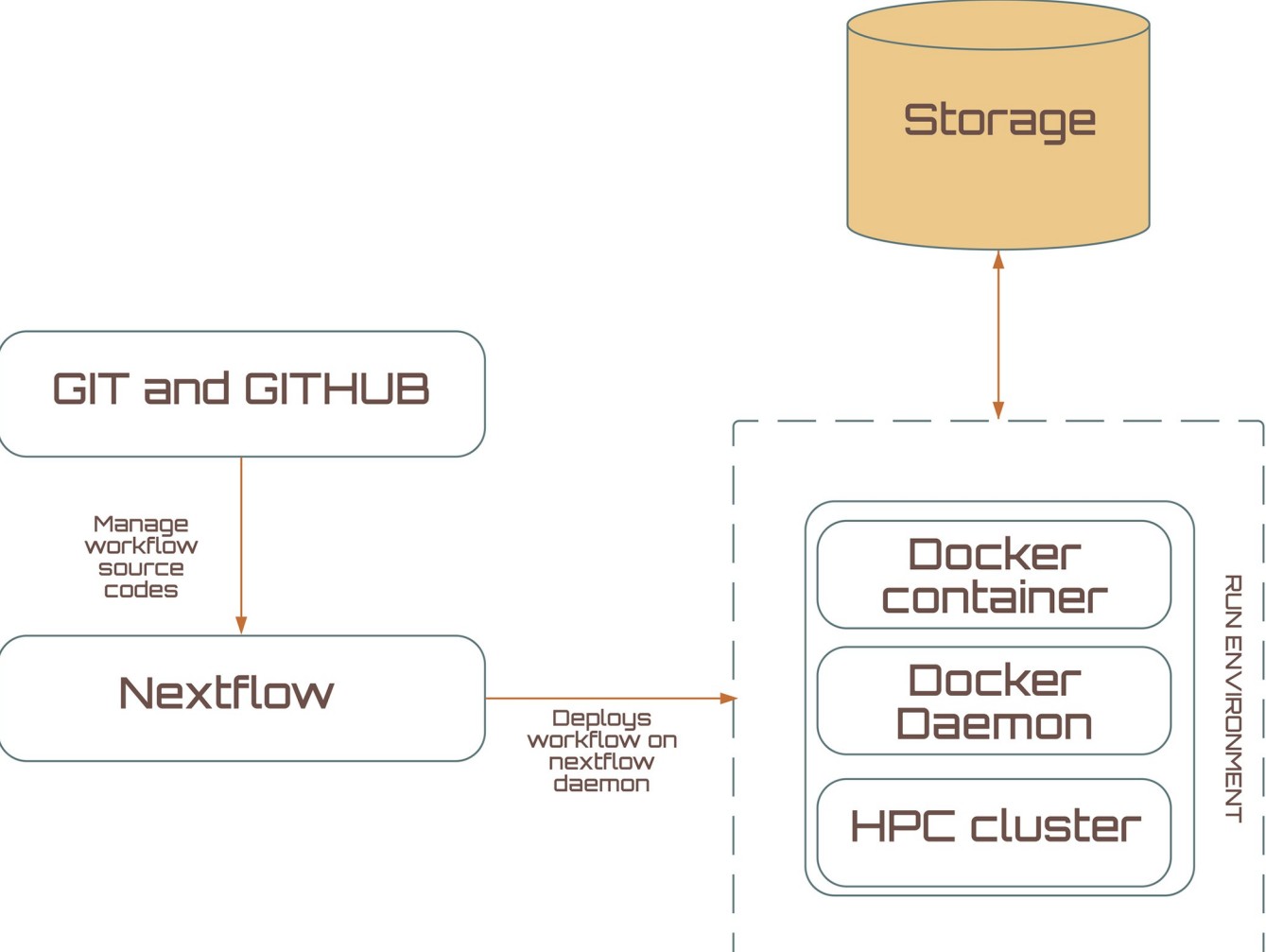

**Fig 2. Overall architecture of the Kuura pipeline.** The source codes for the pipeline are version controlled using git and maintained in github, the run environment is stored as a docker image. Upon initiating the analysis pipeline, nextflow deploys the docker container on top of HPC cluster and runs the analysis within the docker container and upon completion of the process the output is directly written to the specified storage volume.

**Table 3. Summary of the tools and their respective *docker* containers used in each stage.**

| Stage | Tools used | *Docker* container used |
|---|---|---|
| Quality control | *FASTQC* | utuprcagenetics/dnapipe:0.1 |
| | cutadapt/fastp* | |
| | *FASTQC* | |
| Genome alignment and quality score recalibration | BWA-MEM | |
| | *GATK MarkDuplicates* | broadinstitute/*GATK* |
| | *SAMtools* | utuprcagenetics/dnapipe:0.1 |
| | *GATK BaseRecalibrator* | broadinstitute/*GATK* |
| | *GATK* ApplyBQSR | |
| | *Mosdepth* | quay.io/biocontainers/mosdepth:0.2.4—he527e40_0 |
| | *bedtools* | utuprcagenetics/dnapipe:0.1 |
| Variant calling & variant recalibration | *GATK Haplotypecaller* | broadinstitute/*GATK* |
| | *GATK VariantRecalibrator* | |
| | *GATK* ApplyVQSR | |
| | *DeepVariant* | google/*DeepVariant*:1.4.0 |
| | *Strelka*2 | utuprcagenetics/dnapipe:0.1 |
| | *Freebayes* | |
| | *VarScan*2 | |
| Variant consensus & annotation | *BCFtools* | |
| | *VEP* | ensemblorg/ensembl-*VEP* |
| Summary | *MultiQC* | utuprcagenetics/dnapipe:0.1 |

*Cutadapt is used for adaptor trimming if the adaptor sequence is provided, else fastp is used.

## Portability and reproducibility

The pipeline has been constructed to run seamlessly from start to end, it starts with fastq files and produces annotated vcf files that users can directly use for downstream analysis. The overall architecture of the pipeline is illustrated in Fig 2 and the instructions for installation and testing are briefed below.

Kuura uses *nextflow*, a framework for creating scalable bioinformatics pipelines, to enable reproducibility and flexible computational resource management. Git and github are used for version control & collaboration while *docker* is used to preserve and implement the run environment. The list of tools used in each stage are provided in Table 3.

## Setting up and running Kuura pipeline

1. Install and configure *nextflow* & *docker*.

2. Clone the source codes from the github repository.

3. After downloading the github repository, the *docker* image required for running the pipeline needs to be built. Change into the docker directory and run the command '*docker build -t dna-seq-pipeline:0.1.*'

4. Configure the required variables for running the pipeline. In the configuration folder there is a file called 'default.config'where variables such as input data directory, output directory, source data for various processes, data descriptors, etc., need to be completed.

**Fig 3. Screenshot showing a successfully executed pipeline and the information presented while the pipeline is running.**

5. The pipeline is ready to be run and can be started by running *nextflow* with the '-with-*docker*'option. *Nextflow* will run the pipeline in a *docker* container using the specified *docker* image. The command will look like; "*nextflow run bin/Nextflow/dna-seq-pipeline.nf -w <workspace directory> -profile default*"

6. A screenshot of the successfully executed workflow is attached in Fig 3. It shows information on the data used, computational parameters used, steps successfully completed, date and time taken to complete the workflow.

## Supporting information

**S1 File. Detailed installation and usage instructions.**
(DOCX)

**S1 Table. Complete validation results.** In the revision process, the pipeline was validated on gold standard data sets HG003, HG004, HG006 and HG007, data sets generated with the same sequencing protocol in the same study as data sets HG001, HG002 and HG005. The table shows the number of variants identified by each variant caller, their precision and recall values. *The table contains only SNP information.
(XLSX)

## Acknowledgments

Partial development and validation of this pipeline was performed using the ePouta private cloud service provided by CSC–IT CENTER FOR SCIENCE LTD. We are thankful for their support with this work.

## Author Contributions

**Conceptualization:** Dhanaprakash Jambulingam, Venkat Subramaniam Rathinakannan, Samuel Heron, Johanna Schleutker, Vidal Fey.

**Funding acquisition:** Johanna Schleutker.

**Investigation:** Dhanaprakash Jambulingam, Venkat Subramaniam Rathinakannan.

**Methodology:** Dhanaprakash Jambulingam, Venkat Subramaniam Rathinakannan, Samuel Heron, Vidal Fey.

**Project administration:** Johanna Schleutker, Vidal Fey.

**Resources:** Johanna Schleutker, Vidal Fey.

**Software:** Dhanaprakash Jambulingam, Venkat Subramaniam Rathinakannan, Samuel Heron, Vidal Fey.

**Supervision:** Johanna Schleutker, Vidal Fey.

**Validation:** Dhanaprakash Jambulingam, Vidal Fey.

**Writing – original draft:** Dhanaprakash Jambulingam, Venkat Subramaniam Rathinakannan, Vidal Fey.

**Writing – review & editing:** Dhanaprakash Jambulingam, Johanna Schleutker, Vidal Fey.

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
