## [Decision Letter · Decision Letter 0]

17 Jul 2023

PONE-D-23-14875Kuura - An automated workflow for analyzing WES and WGS dataPLOS ONE

Dear Dr. Fey,

Thank you for submitting your manuscript to PLOS ONE. After careful consideration, we feel that it has merit but does not fully meet PLOS ONE’s publication criteria as it currently stands. Therefore, we invite you to submit a revised version of the manuscript that addresses the points raised during the review process.

In particular please address the reviewer comments about making your pipeline easy to install and run. Also please address the comment about the performance 

We look forward to receiving your revised manuscript.

Kind regards,

Nikolas Pontikos

Academic Editor

PLOS ONE

Journal Requirements:

4. We note that Figure 2 in your submission contain copyrighted images. All PLOS content is published under the Creative Commons Attribution License (CC BY 4.0), which means that the manuscript, images, and Supporting Information files will be freely available online, and any third party is permitted to access, download, copy, distribute, and use these materials in any way, even commercially, with proper attribution. For more information, see our copyright guidelines: http://journals.plos.org/plosone/s/licenses-and-copyright.

Additional Editor Comments:

Please address the reviewer comments below especially regarding the performance of the pipeline and its installation so that it can be easily installed, ran and tested.

Reviewers' comments:

Reviewer's Responses to Questions

**Comments to the Author**

1. Is the manuscript technically sound, and do the data support the conclusions?

Reviewer #1: Yes

Reviewer #2: Yes

2. Has the statistical analysis been performed appropriately and rigorously? 

Reviewer #1: Yes

Reviewer #2: N/A

3. Have the authors made all data underlying the findings in their manuscript fully available?

Reviewer #1: Yes

Reviewer #2: Yes

4. Is the manuscript presented in an intelligible fashion and written in standard English?

Reviewer #1: Yes

Reviewer #2: Yes

5. Review Comments to the Author

Reviewer #1: The manuscript provides a Nextflow-based pipeline that bioinformaticians can use in their analysis of genetic variants. The pipeline runs as described but the setting up requires some clarification.

Step 3 in line 154 of manuscript, the command used only works in the docker directory and the change of directory needs to be specified in the step.

Files in the GitHub repository are named differently (default.config to standard.config) and a better guide on how to set up the config files would greatly extend the user-friendliness of this pipeline. It would also be beneficial to add supplementary files the repository explaining what each file is or a README file. Users who have not run Nextflow before would appreciate these comments.

The analysis is limited since it does not compare the effectiveness of Kuura to other pipelines based on variants called, time taken for analysis, processing power, etc. Evidence of this comparison would further convince and guide the implementation of this pipeline to existing workflows.

Reviewer #2: This work decribed a whole genome/exome sequencing workflow (Kuura) that consumes fastq files and produces annotated short genetic variants with multiple standard QC reports. It is written in Nextflow, with Docker enabled. The authors are aware of other pipelines such as SeqMule, DNAp, SpeedSeq and GESALL, but it seems their codebases are no longer maintained (most recent change to them was two years ago to SeqMule). Another Nextflow DnaSeq pipeline is Sarek that is still actively maintained and has a relatively large userbase, however its codebase is heavy and its structure is complex.

In contrast, Kuura provides a light-weight solution that requires a minimal setup and is easy to expand functionalities. It can also serve as a backbone for more advanced workflows.

Major concerns:

1. The data presented in Table 3 are surprising. The precisions and recalls are worryingly low. These numbers should be much higher according to Zhao et al., 2020 (https://doi.org/10.1038/s41598-020-77218-4).

Minor concerns:

1. There is no Readme in the github repository. A Readme with a brief desription of the workflow, and instructions to setup and test is necessary.

2. The existing workflow exclusively supports Docker as a solution. However, Docker's incompatibility with many High Performance Computing (HPC) systems hampers its widespread utilization. The authors should consider to include Singularity. (For reference, https://depot.galaxyproject.org/singularity/ has a wide range of images)

3. There are hardcoded parameters in the workflow (e.g. https://github.com/dhanaprakashj/kuura_pipeline/blob/main/dnaseq/main.nf#L96, https://github.com/dhanaprakashj/kuura_pipeline/blob/main/dnaseq/main.nf#L128, https://github.com/dhanaprakashj/kuura_pipeline/blob/main/dnaseq/main.nf#L206). Please make them configurable.

4. The workflow will work best with a small number of exomes. I think it might struggle with large number of exome samples or whole genomes. The authors should discucss this.

6. PLOS authors have the option to publish the peer review history of their article (what does this mean?). If published, this will include your full peer review and any attached files.

Reviewer #1: No

Reviewer #2: **Yes: **Jing Yu

---

## [Author Response · Author response to Decision Letter 0]

13 Oct 2023

Journal Requirements:

Comment 1. Please ensure that your manuscript meets PLOS ONE's style requirements, including those for file naming. The PLOS ONE style templates can be found at 

Response: Thank you for pointing that out. We have corrected the author’s affiliation information to meet the style requirements.

Comment 2. Please note that PLOS ONE has specific guidelines on code sharing for submissions in which author-generated code underpins the findings in the manuscript. In these cases, all author-generated code must be made available without restrictions upon publication of the work. Please review our guidelines at https://journals.plos.org/plosone/s/materials-and-software-sharing#loc-sharing-code and ensure that your code is shared in a way that follows best practice and facilitates reproducibility and reuse.

Response: Thank you for the comment. We added a section “Code Availability and Accessibility” naming the code repository. All code is made publicly available at https://github.com/dhanaprakashj/kuura_pipeline.

Comment 3. In your Data Availability statement, you have not specified where the minimal data set underlying the results described in your manuscript can be found. PLOS defines a study's minimal data set as the underlying data used to reach the conclusions drawn in the manuscript and any additional data required to replicate the reported study findings in their entirety. All PLOS journals require that the minimal data set be made fully available. For more information about our data policy, please see http://journals.plos.org/plosone/s/data-availability.

Response: Thank you for that comment. We added a section “Data availability” detailing the FTP access to the downloaded files.

Comment 4. We note that Figure 2 in your submission contain copyrighted images. All PLOS content is published under the Creative Commons Attribution License (CC BY 4.0), which means that the manuscript, images, and Supporting Information files will be freely available online, and any third party is permitted to access, download, copy, distribute, and use these materials in any way, even commercially, with proper attribution. For more information, see our copyright guidelines: http://journals.plos.org/plosone/s/licenses-and-copyright.

Response: Thank you for pointing that out. We removed all potentially copyrighted parts of figure 2 and uploaded a new image file.

Reviewers' comments:

Reviewer #1:

Comment 1. Step 3 in line 154 of manuscript, the command used only works in the docker directory and the change of directory needs to be specified in the step.

Response: Thank you for spotting that, it is indeed an important step that has been missed. We added a statement to change the directory after downloading the git repository.

Comment 2. Files in the GitHub repository are named differently (default.config to standard.config) and a better guide on how to set up the config files would greatly extend the user-friendliness of this pipeline. It would also be beneficial to add supplementary files the repository explaining what each file is or a README file. Users who have not run Nextflow before would appreciate these comments.

Response: Thank you for pointing that out. We corrected the file names in the github repository and added a README file documenting the basic setup procedure.

Comment 3. The analysis is limited since it does not compare the effectiveness of Kuura to other pipelines based on variants called, time taken for analysis, processing power, etc. Evidence of this comparison would further convince and guide the implementation of this pipeline to existing workflows.

Response: Thank you for the remark. We are aware of that and have seen such comparisons in the literature. However, in our opinion, a comparison of this kind would be short-lived as the specifications of our workflow as well as those of the compared tools, and computing resources change quickly. It would also not accurately reflect the main objective of Kuura which is to provide reproducible end-to-end variant calling with minimal setup as detailed in the manuscript. That is, we do not aim at a particular performance level but rather usability and practical applicability in a reproducible manner.

Reviewer #2:

Major concerns:

Comment 1. The data presented in Table 3 are surprising. The precisions and recalls are worryingly low. These numbers should be much higher according to Zhao et al., 2020 (https://doi.org/10.1038/s41598-020-77218-4).

Response: Thank you for the comment; the validation values were indeed very low, a fact that we were aware of, but we reported those values nonetheless as they represent the unbiased results of our analysis.

To find the cause for the discrepancy between previously published validation values and remedy the concern we repeated the validation with different settings.

Below are the steps we took to to arrive at the now presented outcome:

- We used a completely different set of sequencing data; the new set was sequenced on an Illumina NovaSeq 6000 system (as compared to a llumina HiSeq 2500 before).

- We used a different targets bed file for coverage analysis and hap.py validation (idt_capture_novogene.grch38.bed) matching the validation set experiment.

NB: The bed file used before also matched the experiment (provided by the Garvan institute along with the read files).

- We ran the pipeline separately for both trimming tools on sample NA12878, using cutadapt in one setting and fastp in the other, both yielding similar results to those seen when using the respective other tool (see table in the Rebuttal Letter).

NB: The originally presented results were generated using cutadapt on the Garvan samples and the results presented now were generated using fastp on the NovaSeq samples.

In addition, we ran several tests manually, i.e., outside the pipeline environment, including:

- Complete manual pipeline workflow rerun of the Garvan sample NIST7035_TAAGGCGA using only DeepVariant resulting in the same validation outcome as in the previous version of the manuscript.

NB: When we tested different target bed files on the Garvan samples we noticed an improvement whenever NOT using the Garvan bed file, with the idt_capture_novogene.grch38.bed giving the best results (see table in the Rebuttal Letter).

- Running the exome sequencing case study presented on the DeepVariant github site (starting from BAM files) resulting in comparable validation values.

- Complete manual pipeline workflow rerun of the DeepVariant case study data (HG003_NA24149_father read files) using only DeepVariant arriving at comparable results as presented in the case study.

In addition, when running the tests we notice that, by mistake, all detected variants had been counted instead of only SNPs. That was corrected so all new tables now contain only SNP counts.

As can be seen in the revised table in the manuscript the new data yielded results much closer to those published in recent pipeline comparison papers and case studies. The outcome of abovementioned comparison runs using different settings and input files led us to the conclusion that the originally used Garvan data set is the main cause for the low validation values.

However, since it is not the main goal of this manuscript to evaluate the performance of different variant callers but rather to present an easy-to-use sequencing data analysis pipeline relying on several widely used, validated calling engines, we did not go to the same length in terms of validation before the first submission, obviously missing the potential flaw in the source data. The successful revalidation, though, proves that the pipeline works and further strengthens the point of our proposed consensus call file with high-confidence variants given that different variant callers yield sometimes dramatically different results and may indeed be not applicable to all types of sequencing projects.

Minor concerns:

Comment 1. There is no Readme in the github repository. A Readme with a brief description of the workflow, and instructions to setup and test is necessary.

Response: Thank you for the comment; the README file was indeed forgotten when creating the remote repository and now added.

Comment 2. The existing workflow exclusively supports Docker as a solution. However, Docker's incompatibility with many High Performance Computing (HPC) systems hampers its widespread utilization. The authors should consider to include Singularity. (For reference, https://depot.galaxyproject.org/singularity/ has a wide range of images)

Response: This is a relevant point, and we are aware of the limitations regarding the use of Docker for security reasons. Docker was used initially because we had the most experience with this container software but have worked on a Singularity container which will be added to the repository in a future release of the pipeline. Singularity is able to pull, build from and run Docker containers which might be a solution for more experienced users already now.

Comment 3. There are hardcoded parameters in the workflow (e.g. https://github.com/dhanaprakashj/kuura_pipeline/blob/main/dnaseq/main.nf#L96, https://github.com/dhanaprakashj/kuura_pipeline/blob/main/dnaseq/main.nf#L128, https://github.com/dhanaprakashj/kuura_pipeline/blob/main/dnaseq/main.nf#L206). Please make them configurable.

Response: Thank you for pointing those out, we corrected these parameters.

Comment 4. The workflow will work best with a small number of exomes. I think it might struggle with large number of exome samples or whole genomes. The authors should discucss this.

Response: Thank you for pointing out that concern. We have tested the pipeline on WES as well as WGS data sets and found, apart from the run time and larger requirements in terms of computing resources, that it performs equally well for both.

---

## [Decision Letter · Decision Letter 1]

19 Dec 2023

Kuura - An automated workflow for analyzing WES and WGS data

PONE-D-23-14875R1

Dear Dr. Fey,

We’re pleased to inform you that your manuscript has been judged scientifically suitable for publication and will be formally accepted for publication once it meets all outstanding technical requirements.

Kind regards,

Alvaro Galli

Academic Editor

PLOS ONE

Additional Editor Comments (optional):

Reviewers' comments:

Reviewer's Responses to Questions

**Comments to the Author**

1. If the authors have adequately addressed your comments raised in a previous round of review and you feel that this manuscript is now acceptable for publication, you may indicate that here to bypass the “Comments to the Author” section, enter your conflict of interest statement in the “Confidential to Editor” section, and submit your "Accept" recommendation.

Reviewer #2: All comments have been addressed

2. Is the manuscript technically sound, and do the data support the conclusions?

Reviewer #2: Yes

3. Has the statistical analysis been performed appropriately and rigorously? 

Reviewer #2: N/A

4. Have the authors made all data underlying the findings in their manuscript fully available?

Reviewer #2: Yes

5. Is the manuscript presented in an intelligible fashion and written in standard English?

Reviewer #2: Yes

6. Review Comments to the Author

Reviewer #2: The authors have addressed all my comments. However, as I ran through it again I think that having the option to include compensating set of variants, i.e. variants called by some variant callers, but not all, could be an important improvement, e.g. for finding rare variants underlying Mendelian disorders.

7. PLOS authors have the option to publish the peer review history of their article (what does this mean?). If published, this will include your full peer review and any attached files.

Reviewer #2: **Yes: **Jing Yu

---

## [Editor Report · Acceptance letter]

9 Jan 2024

PONE-D-23-14875R1 

PLOS ONE

Dear Dr. Fey, 

I'm pleased to inform you that your manuscript has been deemed suitable for publication in PLOS ONE. Congratulations! Your manuscript is now being handed over to our production team.

Kind regards, 

on behalf of

Dr. Alvaro Galli 

Academic Editor

PLOS ONE